# ChePAN: Constrained Black-Box Uncertainty Modelling with Quantile Regression

## Abstract

Most predictive systems currently in use do not report any useful information for auditing their associated uncertainty and evaluating the corresponding risk. Taking it for granted that their replacement may not be advisable in the short term, in this paper we propose a novel approach to modelling confidence in such systems while preserving their predictions. The method is based on the Chebyshev Polynomial Approximation Network (the ChePAN), a new way of modelling aleatoric uncertainty in a regression scenario. In the case addressed here, uncertainty is modelled by building conditional quantiles on top of the original pointwise forecasting system considered as a black box, i.e. without making assumptions about its internal structure. Furthermore, the ChePAN allows users to consistently choose how to constrain any predicted quantile with respect to the original forecaster. Experiments show that the proposed method scales to large size data sets and transfers the advantages of quantile regression to estimating black-box uncertainty.

## 1 Introduction

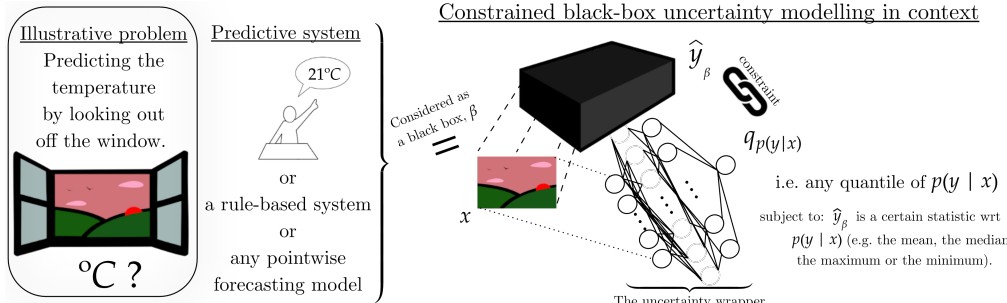

Figure 1: Description of the uncertainty modelling of a black-box predictive system, $\beta$. This modelling is done by means of an uncertainty wrapper (the only part of the ChePAN that requires a neural network), which produces all of the distribution $p(y \mid x)$ as quantiles, $q_{p(y|x)}$. The ChePAN ensures that the original prediction of $\beta$ corresponds to a desired statistic of $p(y \mid x)$, i.e. the constraint.

The present paper proposes a novel method for adding aleatoric uncertainty estimation to any pointwise predictive system currently in use. Considering the system as a *black box*, i.e. avoiding any hypothesis about the internal structure of the system, the method offers a solution to the *technical debt* debate. The concept of *technical debt* was introduced in 1992 to initiate a debate on the long-term costs incurred when moving quickly in software engineering (Sculley et al. (2015); Cunningham (1992)). Specifically, most of the predictive systems currently in use have previously required much effort in terms of code development, documentation writing, unit test implementation, preparing dependencies or even their compliance with the appropriate regulations (e.g., medical (Ustun & Rudin (2016)) or financial models (Rudin (2019)) may have to satisfy interpretability constraints). However, once the system is being used with real-world problems, a new requirement can arise regarding the confidence of its predictions when the cost of an erroneous prediction is high. That being said, replacing the currently-in-use system may not be advisable in the short term. To address this issue,

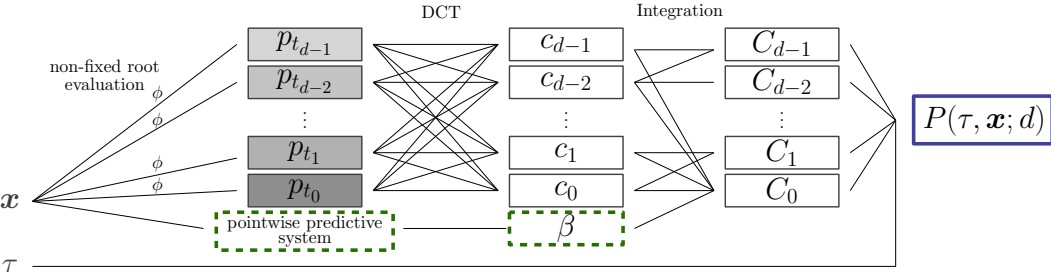

Figure 2: Graphic representation of the ChePAN. For any degree $d$, $\{p_{t_i}\}_{i=0}^{d-1}$ are evaluations of the initial Chebyshev polynomial expansion, $\{c_k\}_{k=0}^{d-1}$ their coefficients, $\{C_k\}_{k=0}^{d-1}$ the coefficients of the integrated polynomial, $\beta$ the black box function and $P$ the conditional prediction of the quantile $\tau$.

the aim of this work is to report any information that is useful for auditing the system's associated uncertainty without modifying its predictions.

In general terms, sources of uncertainty can be understood by analysing the conditional members of this joint distribution: $p(y, \boldsymbol{x}) = \int_{\mathbb{M}} p(y \mid \boldsymbol{x}, M) p(M \mid \boldsymbol{x}) p(\boldsymbol{x}) \, dM$ where $M \in \mathbb{M}$ is the family (assumed non-finite) of models being considered.

Not all methods developed to model uncertainty can be applied in the black-box scenario, since the main hypothesis is that the black box is a fixed single model and unknown internally. Here, we refer specifically to those solutions that model *epistemic* uncertainty, which requires modelling $p(M \mid \boldsymbol{x})$. By epistemic, we mean that uncertainty which can derive from ignorance about the model, including, for example, ensemble models (Lakshminarayanan et al. (2017)), Bayesian neural networks (Rasmussen (1996); Blundell et al. (2015); Hernández-Lobato & Adams (2015b); Teye et al. (2018)) or MC-Dropout (Gal & Ghahramani (2016)).

However, the black box could be a non-parametric predictive system or even a handcrafted rule-based system, as shown in Figure 1. Hence the reason for studying aleatoric uncertainty (Der Kiureghian & Ditlevsen (2009); Kendall & Gal (2017); Brando et al. (2019)), which originates from the variability of possible correct answers given the same input data, $p(y \mid \boldsymbol{x})$. This type of uncertainty can be tackled by modelling the response variable distribution. For instance, imposing a conditional normal distribution where the location parameter is the black-box function and the corresponding scale parameter is learnt. However, the more restricted the assumptions made about this distribution, the more difficult it will be to model heterogeneous distributions. One solution to this limitation is the type of regression analysis used in statistics and econometrics known as *Quantile Regression* (QR), which will provide a more comprehensive estimation.

Unlike classic regression methods, which only estimate a selected statistic such as the mean or the median, QR allows us to approximate any desired quantile. The main advantage of this method is that it allows confidence intervals to be captured without having to make strong assumptions about the distribution function to be approximated.

Recently, several works (Dabney et al. (2018a); Tagasovska & Lopez-Paz (2018); Brando et al. (2019)) have proposed a single deep learning model that implicitly learns all the quantiles at the same time, i.e. the model can be evaluated for any real value $\tau \in [0, 1]$ to give a pointwise estimation of any quantile value of the response variable. Nevertheless, these QR solutions are not directly applicable to the uncertainty modelling of a black box because the predicted quantiles need to be linked to the black-box prediction in some way.

In the present paper, we propose a novel method for QR based on estimating the derivative of the final function using a Chebyshev polynomial approximation to model the uncertainty of a black-box system. Specifically, this method disentangles the estimation of a selected statistic $\beta$ of the distribution $p(y \mid \boldsymbol{x})$ from the estimation of the quantiles of $p(y \mid \boldsymbol{x})$ (shown in Figure 2). Hence, our method is not restricted to scenarios where we can jointly train both estimators, but can also be applied to pre-existing regression systems as a wrapper that produces the necessary information to evaluate aleatoric uncertainty. Additionally, the proposed method scales to several real-world data sets.

This paper is organised as follows. Section 2 states the real-world motivation of the current research as well as the contribution it will be presented. Section 3 introduces the problem of QR and reviews the classic approach to use with neural networks, showing how it cannot be applied directly to constrained black-box uncertainty modelling. Section 4 explores an approach for modelling the derivative of a function using neural networks. The two previous sections provide the baseline for developing our proposed model and its properties, which is presented in Section 5. And finally, in Section 6, we show how our model can be applied in large data sets and defines a new way of modelling the aleatoric uncertainty of a black box. The results are then summarised in the conclusion.

## 2 RESEARCH GOAL AND CONTRIBUTION

The present article was motivated by a real-world need that appears in a pointwise regression forecasting system of a large company. Due to the risk nature of the internal problem where it is applied, uncertainty modelling is important. However, similarly to the medical or financial cases presented in the introduction, interpretability requirements were essential in defining the model currently used by the company, which does not report confidence any prediction made. The need for this research arises in cases where the replacement of the aforementioned system is not advisable in the short term, despite the ongoing need for the uncertainty estimation of that system.

**Definition of constrained black-box uncertainty modelling**

From the probabilistic perspective, solving a regression problem involves determining a conditional density model, $q(y \mid \boldsymbol{x})$. This model fits an observed set of samples $\mathcal{D} = (X, Y) = \left\{ (\boldsymbol{x}_i, y_i) \mid \boldsymbol{x}_i \in \mathbb{R}^D, y_i \in \mathbb{R} \right\}_{i=1}^{n}$, which we assume to be sampled from an unknown distribution $p(y \mid \boldsymbol{x})$. i.e. the real data. Given this context, the pointwise forecasting system mentioned above is a function, $\beta \colon \mathbb{R}^D \to \mathbb{R}$, which tries to approximate a certain conditional summary statistic (a percentile or moment) of $p(y \mid \boldsymbol{x})$.

Regarding the notation, we will call the "constraint" the known or assumed summary statistic that is approximated by $\beta(\boldsymbol{x})$ (e.g. if $\beta$ is reducing the mean square error, then it corresponds to the conditional mean. Otherwise, if it minimises the mean absolute error, it corresponds to the median).

Importantly, in the constrained black-box uncertainty modelling context, the mismatch between the real conditional statistic and the black box, $\beta$, becomes a new source of aleatoric uncertainty that is different from the one derived from the data. However, the way to model it continues to be by estimating $p(y \mid \boldsymbol{x})$. Therefore, a poorly estimated $\beta$ will impact the modelling of $p(y \mid x)$, given that we always force the constraint to be satisfied (as shown in Figure 3 of the Experiment section).

So far, we have attempted to highlight the fact that we do not have a strong hypothesis about the internals of this $\beta$ function, we have only assumed that it approximates a certain statistic of $p(y \mid \boldsymbol{x})$. Accordingly, we call this function the "constrained black box". This flexible assumption will enable us to consider several pointwise models as $\beta$, as shown in Figure 1.

The overall goal of the present article is, taking a pre-defined black box $\beta(\boldsymbol{x})$ that estimates a certain conditional summary statistic of $p(y \mid \boldsymbol{x})$, to model $q(y \mid \boldsymbol{x})$ under the constraint that if we calculate the summary statistic of this predicted conditional distribution, it will correspond to $\beta(\boldsymbol{x})$.

As mentioned in the Introduction, since we have a fixed black box, we are unable to apply Bayesian techniques such as those that infer the distribution of parameters within the model, $p(M \mid \boldsymbol{x})$. In general, even though they are very common techniques in generic uncertainty modelling, no such epistemic uncertainty techniques can be applied in this context due to the limitation of only having a single fixed model.

In addition, it should be noted that not all models that estimate $p(y \mid \boldsymbol{x})$ can be used in the constrained black-box uncertainty modelling context. To solve this problem, we require models that predict $q(y \mid \boldsymbol{x})$ but also force the chosen conditional summary statistic of $q(y \mid \boldsymbol{x})$ to have the same value as $\beta(\boldsymbol{x})$. The main contribution of this work is to present a new approach that allows us not only to outperform other baseline models when tackling this problem, but also to decide which kind of constraint we wish to impose between $\beta(\boldsymbol{x})$ and $q(y \mid \boldsymbol{x})$. The $q(y \mid \boldsymbol{x})$ will be approximated using Quantile Regression (explained in Section 3) and the constraint will be created considering the integration constant of the $q(y \mid \boldsymbol{x})$ derivative (shown in Section 5.1).

## 3    CONDITIONAL QUANTILE REGRESSION

In *Quantile Regression* (QR), we estimate $q$ in a discrete manner by means of quantiles, which does not assume any typical parametric family distribution to the predicted $p$, i.e. it goes beyond central tendency or unimodality assumptions.

For each quantile value $\tau \in [0, 1]$ and each input value $\boldsymbol{x} \in \mathbb{R}^D$, the conditional quantile function will be $f \colon [0, 1] \times \mathbb{R}^D \to \mathbb{R}$. In our case, we use deep learning as a generic function approximator (Hornik et al. (1989)) to build the model $f$, as we shall see later. Consequently, $f$ is a parametric function that will be optimised by minimising the following loss function with respect to their weights $\boldsymbol{w}$,

$$\mathcal{L}(\boldsymbol{x}, y, \tau) = \big(y - f_{\boldsymbol{w}}(\tau, \boldsymbol{x})\big) \cdot \big(\tau - \mathbb{1}[y < f_{\boldsymbol{w}}(\tau, \boldsymbol{x})]\big) \tag{1}$$

where $\mathbb{1}[c]$ denotes the indicator function that verifies the condition $c$. Equation 1 is an asymmetric convex loss function that penalises overestimation errors with weight $\tau$ and underestimation errors with weight $1 - \tau$.

Recently, different works (Dabney et al. (2018b;a); Wen et al. (2017)) have proposed deep learning models that minimise a QR loss function similar to Equation 1. For instance, in the field of reinforcement learning, the Implicit Quantile Network (IQN) model was proposed (Dabney et al. (2018a)) and subsequently applied to solve regression problems as the Simultaneous Quantile Regression (SQR) model (Tagasovska & Lopez-Paz (2019)) or the IQN in (Brando et al. (2019)). These models consist of a neural network $\psi \colon [0, 1] \times \mathbb{R}^D \to \mathbb{R}$ such that it directly learns the function $f$ that minimises Equation 1, i.e. $f = \psi$. In order to optimise $\psi$ for all possible $\tau$ values, these models pair up each input $\boldsymbol{x}$ with a sampled $\tau \sim \mathcal{U}(0, 1)$ from a uniform distribution in each iteration of the stochastic gradient descent method. Thus, the final loss function is an expectation over $\tau$ of Equation 1.

However, these QR models **cannot be applied to the constrained black-box scenario**, given that they do not link their predicted quantiles with a pointwise forecasting system in a constrained way (Section 5.1). Other models, such as quantile forests, have a similar limitation. In the next section, we introduce the other main part required to define our proposed method.

## 4    MODELLING THE DERIVATIVE WITH A NEURAL NETWORK

Recently, a non-QR approach was proposed to build a monotonic function based on deep learning: the Unconstrained Monotonic Neural Network (UMNN) (Wehenkel & Louppe (2019)). The UMNN estimates the derivative of a function by means of a neural network, $\phi$, which has its output restricted to strictly positive values, i.e. approaching $H(z)$ such that

$$H(z) = \int_0^z \phi(t) \, dt + H(0). \tag{2}$$

Therefore, if the neural network $\phi(z) \approx \frac{\partial H}{\partial z}(z) > 0$, this is in fact a sufficient condition to force $H(z)$ to be monotone.

To compute the integral of $\frac{\partial H}{\partial z}$, the UMNN approximates the integral of Equation 2 using the Clenshaw-Curtis quadrature, which has a closed expression. The UMNN is designed to obtain a general monotonic function with respect to all the model inputs, $z$, but our interest is to build a partial monotonic function with respect to the quantile value, as we will explain hereafter.

The partial monotonic function will be obtained using the Clenshaw-Curtis Network (CCN) model, which is an extension of the UMNN model introduced in Section A.3 of the Appendix and an intermediate step we took to arrive at the main proposal of the current article. Importantly, we have not included it in the main article because it cannot be applied to the constrained black-box uncertainty modelling scenario (as described in Section A.3).

## 5    CHEPAN: THE CHEBYSHEV POLYNOMIAL APPROXIMATION NETWORK

In this section, we will extend the UMNN to a model that is only monotonic with respect to the quantile input $\tau$. Moreover, we will exploit the fact that the quantile domain is in $[0, 1]$ to provide

an approach which is uniformly defined over all of the interval. We call this approach the Chebyshev Polynomial Approximation Network (ChePAN), which allows us to transfer the advantages of quantile regression to the constrained uncertainty modelling of a black box.

As Figure 2 shows, the ChePAN contains a neural network $\phi\colon [0,1] \times \mathbb{R}^D \to \mathbb{R}_+$ that only produces positive outputs and models the derivative of the final function with respect to $\tau$. The goal is to optimise the neural networks $\phi(\tau, \boldsymbol{x})$ by calculating the coefficients of a truncated Chebyshev polynomial expansion $p(\tau, \boldsymbol{x}; d)$ of degree $d$ with respect to $\tau$. That is, we will use a Chebyshev polynomial (described in Section A.1 of the Appendix) to give a representation of the neural network, $\phi$, uniformly defined in $\tau \in [0,1]$. After that, we will use its properties to model the uncertainty of a black box in a constrained way (described in Section 5.1).

Internally, the ChePAN considers a finite mesh of quantile values, called *Chebyshev roots*, $\{t_k\}_{k=0}^{d-1} \subset [0,1]$ and defined by

$$t_k := \frac{1}{2}\cos\frac{\pi(k + \frac{1}{2})}{d} + \frac{1}{2}, \quad 0 \leqslant k < d. \tag{3}$$

The truncated Chebyshev expansion of a function can be interpreted as a linear transformation using a set of evaluations of $\phi$ at the roots, i.e. $\{\phi(t_k, \boldsymbol{x})\}_{k=0}^{d-1}$. This linear transformation gives a vector of coefficients, which are known as Chebyshev coefficients and depend on $\boldsymbol{x}$, i.e. $\{c_k(\boldsymbol{x})\}_{k=0}^{d-1}$, as illustrated in Figure 2.

The implementation of a linear transformation generally has a square complexity. However, the transformation involved in Chebyshev coefficients can be computed efficiently with a $\Theta(d \log d)$ complexity. In fact, the algorithm that speeds the computation is based on the Fast Fourier Transform (FFT) and known as the Discrete Cosine Transform of type-II (DCT-II) (discussed in Section A.1 of the Appendix).

Once the Chebyshev coefficients $c_k(\boldsymbol{x})$ have been computed, we can write them in a linear combination of Chebyshev polynomials $T_k(t)$, i.e.

$$p(\tau, \boldsymbol{x}; d) := \frac{1}{2}c_0(\boldsymbol{x}) + \sum_{k=1}^{d-1} c_k(\boldsymbol{x})T_k(2\tau - 1), \tag{4}$$

where $T_k(t)$ are defined recurrently as $T_0(t) = 1$, $T_1(t) = t$, and $T_{k+1}(t) = 2tT_k(t) - T_{k-1}(t)$ for $k \geqslant 1$. These polynomials $T_k$ do not need to be explicitly computed to evaluate $p$ on a quantile (Clenshaw (1955)).

Note that, given the construction of the coefficients $c_k(\boldsymbol{x})$, the $p(t_k, \boldsymbol{x}; d)$ is equal to $\phi(t_k, \boldsymbol{x})$ at each of the root points $t_k$. These equalities must be understood in terms of machine precision in the numerical representation system, classically $\sim 10^{-16}$ in double-precision or $\sim 10^{-8}$ in single-precision arithmetic. In Figure 2, we denote this root evaluation step as $p_{t_k}$.

The final goal is to provide $P(\tau, \boldsymbol{x}; d)$ so that it approximates the integral of $p$, that is $\int_0^\tau p(t, \boldsymbol{x}; d)\, dt$. Specifically, the integral will also be the integral of the neural network $\phi$,

$$P(\tau, \boldsymbol{x}; d) \approx \Phi(\tau, \boldsymbol{x}) = \int_0^\tau \phi(t, \boldsymbol{x})\, dt + K(\boldsymbol{x}). \tag{5}$$

Since $\phi(\tau, \boldsymbol{x})$ is defined as positive for all $\tau \in [0,1]$, then $P(\tau, \boldsymbol{x}; d)$ will be an increasing function with respect to $\tau$.

Additionally, given that $p(\tau, \boldsymbol{x}; d)$ is a Chebyshev polynomial (defined in Equation 4), its integral w.r.t. $\tau$ is simply the integral of the Chebyshev polynomial $T_k$, which corresponds to a new Chebyshev polynomial. Using the recurrent definition of $T_k$, we deduce the indefinite integrals

$$\int T_0(t)\, dt = T_1(t), \quad \int T_1(t)\, dt = \frac{T_2(t)}{4} - \frac{T_0(t)}{4}, \quad \int T_k(t)\, dt = \frac{T_{k-1}(t)}{2(k-1)} - \frac{T_{k+1}(t)}{2(k+1)}, \tag{6}$$

which leads to the conclusion that $P$ can be given in terms of Chebyshev coefficients as well. Thus,

$$P(\tau, \boldsymbol{x}; d) := \frac{1}{2}C_0(\boldsymbol{x}) + \sum_{k=1}^{d-1} C_k(\boldsymbol{x})T_k(2\tau - 1), \tag{7}$$

where the coefficients $C_k(\boldsymbol{x})$ have a recurrent expression in terms of a Toeplitz matrix (see Clenshaw (1955)). Indeed, by ordering the coefficients of the integral in Equation 4, we deduce that

$$C_k(\boldsymbol{x}) \coloneqq \frac{c_{k-1}(\boldsymbol{x}) - c_{k+1}(\boldsymbol{x})}{4k}, \quad 0 < k < d-1, \quad C_{d-1}(\boldsymbol{x}) \coloneqq \frac{c_{d-2}(\boldsymbol{x})}{4(d-1)}, \tag{8}$$

and $C_0(\boldsymbol{x})$ depends on the constant of integration $K(\boldsymbol{x})$ in Equation 5 and the other coefficient values in Equation 7. This freedom of the predicted $\tau$ in $C_0(\boldsymbol{x})$ allows us to impose a new condition, which becomes a uniform condition in all of the intervals $[0, 1]$. In Section 5.1, we will discuss how to define the $C_0(\boldsymbol{x})$ depending on the black box desired.

## 5.1 Adding an Uncertainty Estimation to a Black-Box Prediction System

In this subsection, we tackle the constrained black-box uncertainty modelling problem introduced in Section 2. The main assumption is that we have a pointwise predictive system, which we will refer to as $\beta(\boldsymbol{x})$ and approximates a **desired** statistic such as the mean, median or a certain quantile of $p(y \mid \boldsymbol{x})$, as shown in Figure 1. It is not necessary for this system to be a deep learning model or even parametric. All that the ChePAN requires to train its neural network, $\phi$, are the corresponding $\beta$-evaluation values of the training set, i.e. $\{\boldsymbol{x}, \beta(\boldsymbol{x})\}$. Thus, the ChePAN calculates the conditioned response distribution to the input without assuming asymmetry or unimodality with respect to this distribution, as well as associating the **desired** statistic of this distribution to $\beta(\boldsymbol{x})$.

The formula used to calculate the constant of integration, $C_0(\boldsymbol{x})$, will depend on which statistic we choose[1]. If we impose the quantile $\tau = 0$ to be $\beta$ (which we shall call *ChePAN-$\beta$=$q_0$*), then

$$C_0(\boldsymbol{x}) = 2\beta(\boldsymbol{x}) - 2\sum_{k=1}^{d-1} C_k(\boldsymbol{x})(-1)^k. \tag{9}$$

However, if we force the quantile $\tau = 1$ to be the $\beta$ (which we shall call *ChePAN-$\beta$=$q_1$*), then

$$C_0(\boldsymbol{x}) = 2\beta(\boldsymbol{x}) - 2\sum_{k=1}^{d-1} C_k(\boldsymbol{x}). \tag{10}$$

For instance, the prediction of extreme weather events involves the forecasting system to predict the maximum or minimum values of $p(y \mid \boldsymbol{x})$. In these cases, this pre-trained system could be used as $\beta$ in Equation 9 or Equation 10, respectively, to determine the overall quantile distribution of $p(y \mid \boldsymbol{x})$, taking $\beta$ as a reference point.

If the median (equivalently, $\tau = 0.5$) is the $\beta$ (which we shall call *ChePAN-$\beta$=Med*), then

$$C_0(\boldsymbol{x}) = 2\beta(\boldsymbol{x}) - 2\sum_{\substack{k=1 \\ k \text{ even}}}^{d-1} (-1)^{k/2} C_k(\boldsymbol{x}). \tag{11}$$

Finally, the mean is forced to be the $\beta$ (which we shall call *ChePAN-$\beta$=Mean*), then

$$C_0(\boldsymbol{x}) = 2\beta(\boldsymbol{x}) - 2\sum_{\substack{k=1 \\ k \text{ odd}}}^{d-1} \frac{C_k(\boldsymbol{x})}{k^2 - 4}. \tag{12}$$

Additionally, $\beta(\boldsymbol{x})$ can be approximated by means of another neural network, which can be simultaneously optimised with $\phi(\tau, \boldsymbol{x})$. We will use this approach to compare the ChePAN and other baseline models in the results section regarding black-box modelling.

## 6 Experiments

The source code used to reproduce the results of the ChePAN in the following experiments can be found in the Github repository[2]. The DCT-II method referred to in Section 5 was used in the aforementioned source code.

---

[1] All details of how such formulas are reached can be found in the supplementary material.

[2] The camera-ready version of this paper will include all of the source codes to reproduce the experiments.

Table 1: Mean and standard deviation of the QR loss value, mean ± std, of 10 executions for each | Black box |-*Uncertainty wrapper* using all of the test distributions in Figure 3 and three data sets (described in Section A.6). The ranges that overlap with the best range are highlighted in bold.

| | Asymmetric | Symmetric | Uniform | Multimodal | Year-MSD | BCN-RPF | YVC-RPF |
|---|---|---|---|---|---|---|---|
| *RF*-N | $42.37 \pm 0.04$ | $23.19 \pm 1.00$ | $66.44 \pm 0.26$ | $151.51 \pm 0.24$ | $57.50 \pm .05$ | $23.47 \pm .14$ | $\mathbf{27.27 \pm .39}$ |
| *RF*-LP | $42.88 \pm 0.04$ | $\mathbf{22.10 \pm 0.03}$ | $67.13 \pm 0.09$ | $153.06 \pm 0.22$ | $57.58 \pm .02$ | $\mathbf{23.07 \pm .17}$ | $28.06 \pm .12$ |
| *RF*-ChePAN | $\mathbf{41.52 \pm 0.35}$ | $23.19 \pm 0.70$ | $\mathbf{65.98 \pm 0.20}$ | $\mathbf{148.39 \pm 0.16}$ | $\mathbf{48.28 \pm .18}$ | $\mathbf{23.17 \pm .07}$ | $28.16 \pm .14$ |
| *XGBoost*-N | $\mathbf{42.42 \pm 0.05}$ | $\mathbf{23.35 \pm 0.99}$ | $66.38 \pm 0.26$ | $149.35 \pm 0.40$ | $51.17 \pm .08$ | $24.52 \pm .26$ | $27.79 \pm .08$ |
| *XGBoost*-LP | $42.90 \pm 0.02$ | $\mathbf{23.02 \pm 0.43}$ | $67.13 \pm 0.17$ | $150.94 \pm 0.12$ | $51.24 \pm .02$ | $22.63 \pm .11$ | $27.86 \pm .07$ |
| *XGB.*-ChePAN | $\mathbf{41.95 \pm 0.40}$ | $\mathbf{23.69 \pm 0.68}$ | $\mathbf{65.89 \pm 0.17}$ | $\mathbf{146.20 \pm 0.30}$ | $\mathbf{48.54 \pm .08}$ | $\mathbf{22.00 \pm .04}$ | $\mathbf{27.51 \pm .13}$ |
| N | $43.63 \pm 2.89$ | $\mathbf{23.70 \pm 6.85}$ | $\mathbf{67.45 \pm 1.68}$ | $148.78 \pm 2.88$ | $49.00 \pm .24$ | $\mathbf{27.28 \pm 1.25}$ | $\mathbf{28.62 \pm 1.61}$ |
| LP | $43.46 \pm 0.15$ | $\mathbf{20.72 \pm 0.47}$ | $\mathbf{68.06 \pm 0.82}$ | $149.99 \pm 0.64$ | $48.67 \pm .28$ | $23.51 \pm .28$ | $22.32 \pm .06$ |
| ChePAN | $\mathbf{41.72 \pm 0.24}$ | $22.94 \pm 1.81$ | $\mathbf{68.55 \pm 6.61}$ | $\mathbf{145.93 \pm 3.14}$ | $\mathbf{46.76 \pm .25}$ | $\mathbf{20.67 \pm .40}$ | $21.97 \pm .12$ |

In this section, we describe the performance of the proposed models compared to other baselines. The main goal is to show that by using QR the ChePAN is an improvement on other black-box uncertainty modelling baselines because it avoids centrality or unimodality assumptions, while also allowing users to choose how to constrain the predicted quantiles with respect to the black-box prediction.

## 6.1 MODELS UNDER EVALUATION

Exponential power distributions satisfy the condition that one of the parameters corresponds to the mode. Thus, those models that approximate such parametric distributions where the mode parameter is the black-box function and estimate the other parameter related to uncertainty can be used as baselines.

- **The Heteroscedastic Normal distribution (N)** Similarly to (Bishop (1994); Kendall & Gal (2017); Tagasovska & Lopez-Paz (2019); Brando et al. (2019)), two neural networks, $\mu$ and $\sigma$, can be used to approximate the conditional normal distribution, $\mathcal{N}(\mu(\boldsymbol{x}), \sigma(\boldsymbol{x}))$, such that they maximise the likelihood.

  In the black-box scenario proposed here, $\mu$ is the black-box function and we only need to optimise the $\sigma$ neural network. Once optimised, the desired quantile $\tau$ can be obtained with $F(\tau, \boldsymbol{x}) = \mu(\boldsymbol{x}) + \sigma(\boldsymbol{x})\sqrt{2} \cdot \text{erf}^{-1}(2\tau - 1)$, $\tau \in (0, 1)$, where $\text{erf}^{-1}$ is the inverse error function.

- **The Heteroscedastic Laplace distribution (LP)** As a more robust alternative to outlier values, a conditional Laplace distribution, $LP\big(\mu(\boldsymbol{x}), b(\boldsymbol{x})\big)$, can be considered. Here, the quantile function is $F(\tau, \boldsymbol{x}) = \mu(\boldsymbol{x}) + \big(b\log(2\tau)\big) \cdot \mathbb{1}\big[\tau \leqslant \frac{1}{2}\big] - \big(b\log(2 - 2\tau)\big) \cdot \mathbb{1}\big[\tau > \frac{1}{2}\big]$, $\tau \in (0, 1)$.

- **The Chebyshev Polynomial Approximation Network (ChePAN)** In order to use the same black boxes as the other baselines, Equation 12 is considered, given that these black boxes are optimising the mean square error. Other alternative equations are considered in the pseudo code and in Figure 6 of the supplementary material.

## 6.2 DATA SETS AND EXPERIMENT SETTINGS

All experiments were implemented in TensorFlow (Abadi et al. (2015)) and Keras (Chollet et al. (2019)), running in a workstation with Titan X (Pascal) GPU and GeForce RTX 2080 GPU. All the details of the data sets used and model hyper-parameters for the results section are described in the supplementary material.

## 6.3 RESULTS

Table 1 shows a comparison of uncertainty modelled for two given black-box systems (a Random Forest (RF) (Liaw et al. (2002)) and an XGBoost (Chen & Guestrin (2016))) in four data sets. The

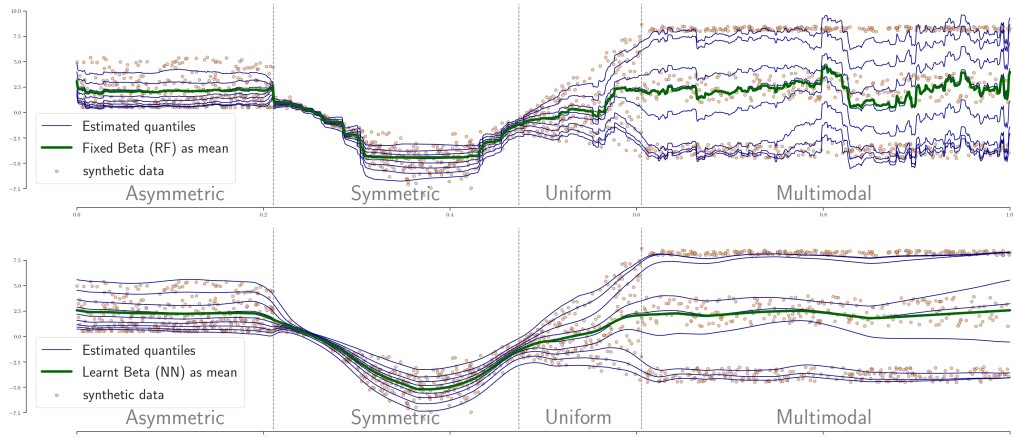

Figure 3: Heterogeneous synthetic distribution proposed by (Brando et al. (2019)). In the upper part of the figure, the learnt quantiles, $\phi$, are noisy because their mean is the black box defined as an inaccurate MSE Random Forest (RF), $\beta$, following Equation 12. In the lower part, $\phi$ and $\beta$ are learnt and asymmetries and multimodalities can be seen more clearly, while still respecting the constraint in Equation 12.

first four columns correspond to each part of the synthetic distribution proposed by (Brando et al. (2019)) and shown in Figure 3, the fifth column is the full Year Prediction MSD UCI dataset (Dua & Graff (2017a)), predicting the release year of a song from 90 audio features and, finally, the last two columns correspond to predicting the room price forecasting of Airbnb flats (RPF) in Barcelona and Vancouver, extracted from (Brando et al. (2019)). The mean of the QR loss value (see Equation 1) is evaluated for ten thousand randomly selected quantiles for ten executions of each model $\{m_k\}_{k=1}^{10}$,

$$\mathcal{L}_{m_k}(X_{test}, Y_{test}) = \sum_{i=1}^{N_{test}} \sum_{j=1}^{N_\tau} \frac{\left(y_i - f_{m_k}(\tau_j, \boldsymbol{x}_i)\right) \cdot \left(\tau_j - \mathbb{1}[y_i < f_{m_k}(\tau_j, \boldsymbol{x}_i)]\right)}{N_{test} \cdot N_\tau}, \qquad (13)$$

where $N_{test}$ is the number of points in the test set, $N_\tau = 10,000$ the number of Monte Carlo samplings and $f_{m_k}$ any of the models considered in Table 1. Considering how the QR loss is defined in Equation 1, its value not only informs us about each system's performance but also how generically calibrated its predicted quantiles are.

Furthermore, in Table 1 we observe that the ChePAN outperforms other methods in most cases due to it transferring the capacity to capture asymmetries and multimodalities of QR in $p(y \mid \boldsymbol{x})$ to the black-box problem, where our uncertainty modelling needs to be restricted in order to maintain the corresponding statistic associated with the black box.

This restriction of conserving the black box can be seen qualitatively in the upper part of Figure 3, where such a restriction must be met in any situation, i.e. even if performance worsens because the black box, $\beta(x)$, is not correctly fitted (as described in Section 2). In this case, $\beta(x)$ is an inaccurate Random Forest predicting the mean. Importantly, the ChePAN propagates the $\beta(x)$ noise to the predicted quantiles (in blue) because the constraint is always forced. On the other hand, the ability of ChePAN to model heterogeneous distributions using QR is better displayed in the lower part of Figure 3. In this case, the black box is a neural network that is learnt concurrently with the quantiles. Since the black box is better approximated, the quantiles are better.

Finally, since Table 1 shows that there is a similar performance order between the baselines when using the RF or XGBoost, we also want to show additional experiments that directly measure the calibration of the predicted quantiles and compare the predicted width of certain desired intervals. Following the UCI data sets used in (Hernández-Lobato & Adams (2015b); Gal & Ghahramani (2016); Lakshminarayanan et al. (2017); Tagasovska & Lopez-Paz (2019)), we performed two empirical studies to assess this point in a black-box scenario where the black box is an MSE-XGBoost. Following the proposed hidden layers architecture in (Tagasovska & Lopez-Paz (2019)), the Prediction Interval Coverage Probability (PICP) and the Mean Prediction Interval Width (MPIW) are reported in Table 3 of the appendix considering the 0.025 and the 0.975 quantiles. For the sake of

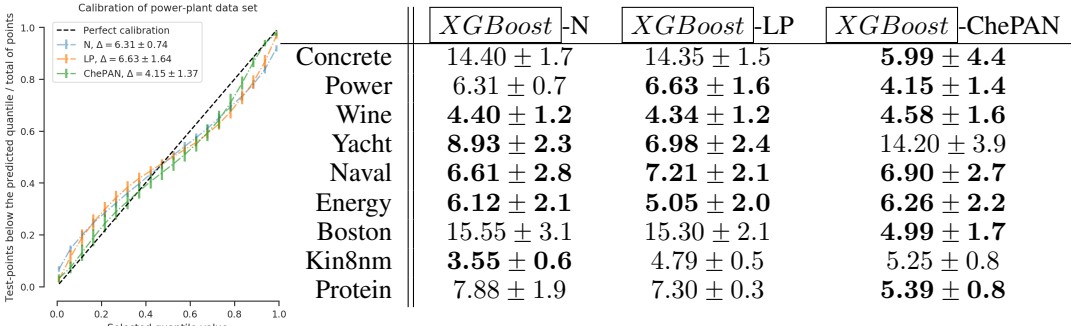

| | $XGBoost$-N | $XGBoost$-LP | $XGBoost$-ChePAN |
|---|---|---|---|
| Concrete | $14.40 \pm 1.7$ | $14.35 \pm 1.5$ | $\mathbf{5.99 \pm 4.4}$ |
| Power | $6.31 \pm 0.7$ | $\mathbf{6.63 \pm 1.6}$ | $\mathbf{4.15 \pm 1.4}$ |
| Wine | $\mathbf{4.40 \pm 1.2}$ | $4.34 \pm 1.2$ | $4.58 \pm 1.6$ |
| Yacht | $\mathbf{8.93 \pm 2.3}$ | $\mathbf{6.98 \pm 2.4}$ | $14.20 \pm 3.9$ |
| Naval | $\mathbf{6.61 \pm 2.8}$ | $\mathbf{7.21 \pm 2.1}$ | $\mathbf{6.90 \pm 2.7}$ |
| Energy | $\mathbf{6.12 \pm 2.1}$ | $\mathbf{5.05 \pm 2.0}$ | $6.26 \pm 2.2$ |
| Boston | $15.55 \pm 3.1$ | $15.30 \pm 2.1$ | $\mathbf{4.99 \pm 1.7}$ |
| Kin8nm | $\mathbf{3.55 \pm 0.6}$ | $4.79 \pm 0.5$ | $5.25 \pm 0.8$ |
| Protein | $7.88 \pm 1.9$ | $7.30 \pm 0.3$ | $\mathbf{5.39 \pm 0.8}$ |

Figure 4: Plot with performance in terms of calibration. The table contains the mean and standard deviation of all the folds using the mean absolute error between the empirical predicted calibration and the perfect ideal calibration of 980 equidistant quantiles using Equation 14.

completeness, in Figure 4 and its associated table we have also computed an additional metric not only to verify the calibration of the 0.025 and 0.975 quantiles, but also to obtain a measure of general calibration considering the entire quantile distribution. Given $N_\tau$-equidistant set of quantiles to evaluate, $\boldsymbol{\tau} = [10^{-2}, \ldots, 1 - 10^{-2}]$, the % of actual test data that falls into each predicted quantile can be compared to each real quantile value as follows,

$$\mathcal{C}al(f; X_{test}, Y_{test}, \boldsymbol{\tau}) = \frac{1}{N_\tau} \sum_{j=1}^{N_\tau} |\tau_j - \frac{1}{N_{test}} \sum_{i=1}^{N_{test}} \mathbb{1}[y_i < f(\tau_j, \boldsymbol{x}_i)]| \tag{14}$$

In addition, two extra figures showing the disentangled visualisation of this calibration metric from each quantile can be found in Figure 5 of the Appendix. As all of the figures and tables show, in terms of calibration, the ChePAN generally displays a better performance in the black-box scenario than the other models.

## 7 CONCLUSION

The uncertainty modelling of a black-box predictive system requires the designing of wrapper solutions that avoid assumptions about the internal structure of the system. Specifically, this could be a non-deep learning model (such as the one presented in Table 1 and Figure 3) or even a non-parametric predictive system, as proposed in Figure 1. Therefore, not all models or types of uncertainties can be considered using this framework.

The present paper introduces the Chebyshev Polynomial Approximation Network (ChePAN) model, which is based on Chebyshev polynomials and deep learning models and has a dual purpose: firstly, it predicts the aleatoric uncertainty of any pointwise predictive system; and secondly, it respects the statistic predicted by the pointwise system.

To conclude, then, the ChePAN transfers the advantages of Quantile Regression (QR) to the problem of modelling aleatoric uncertainty estimation in another existing and fixed pointwise predictive system (denoted as $\beta$ and referred to as a black box). Experiments using different large-scale real data sets and a synthetic one that contains several heterogeneous distributions confirm these novel features.

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
