# OpenReview forum: "ChePAN: Constrained Black-Box Uncertainty Modelling with Quantile Regression"
_ICLR.cc/2021/Conference — Reject_

### Official Review · AnonReviewer2 · 2020-10-28
**Confusing**

**Rating:** 2
**Confidence:** 2

**Review:**

## Summary / Weaknesses

I have to confess that I found the paper to be extremely confusing. It is clear that the submission is still several edits away from a version that can be published.

As far as I can see, the paper lacks a clear, formal description of the problem, and as a result, it is difficult to see precisely where the existing methods fall short (or even which ones are relevant), and how the proposed procedure addresses these shortcomings. Again, as far as I can see, there is little to no theoretical justification for the proposed method in the paper; the proposed method appears to be a heuristic based on approximations that may be reasonable, but requires some justification.

---
## Recommendation
I recommend a reject.

---
## Questions
Please give a formal description of the problem the proposed method is trying to address. Also, please give brief but clear descriptions of the existing approaches and their shortcomings.

I am particularly baffled by the sentence "[the QR models] cannot be be applied to the constrained black-box scenario given that they does (sic) not link their predicted quantiles with a pointwise forecasting system in a constrained way." The way I interpret this sentence is that the existing methods do not model the quantiles (or some other functional) of the _output_ of some pointwise forecasting system, but rather they model functionals of Y | X. If this is the correct interpretation, then I do not see why it would be of interest to model the predictions of a black-box forecasting system rather than Y | X.

I am also confused by the experimental setup. Why do N and LP make sense as comparison methods?

Also, there are several instances of errors of grammar / syntax.

---
## Disclaimer
I did not have the time to look at the supplement.

---
## Update
I thank the authors for responding to my questions and revising the paper.

As indicated in the first sentence of my review, the submission's biggest flaw is in poor presentation. I have read the revised version, but I am afraid to say that I remain disappointed. Although I can see that the authors have tried to address some of the common concerns, the changes do not go far enough. Given that the current version is a version in which "all [...] comments and suggestions have been taken into account," I do not see a reason to give the authors a benefit of the doubt, and have decided to keep my score as is. However, because my evaluation appears to be quite different from those of the other reviewers, I think it may be possible that they are seeing something in this paper that I am incapable of. Since I cannot profess to have a complete understanding of the content, I have decided to downgrade my confidence score.

I will give two examples of ambiguous / confusing language. They are both from Section 2 after the paragraph heading **Definition of constrained black-box uncertainty modeling**.

1. (The last sentence of the 1st paragraph) "Given this context, the pointwise forecasting system mentioned above is a function $\beta:\mathbb{R}^D \to \mathbb{R}$, which tries to approximate a certain conditional summary statistic (a percentile or moment) or $p(y | x)$.

- My understanding of this sentence is that $\beta$ is a _statistic_ (i.e., a function of $\mathcal{D} = (\mathbf{x}_i,y_i)_{i=1}^{n}$) such that $\beta \approx \beta^*$, where $\beta^*$ is some functional of interest of the conditional distribution $p(y | \mathbf{x})$. For example, $\beta^*$ could be the conditional mean $\beta^*(\mathbf{x}) = \mathbb{E}[Y | X = \mathbf{x}]$, in which case $\beta$ is usually obtained by minimizing the MSE, or $\beta^*$ may be the conditional $\alpha$-quantile, in which case $\beta$ is some data approximation of the conditional $\alpha$-quantile.
- Now, compare the above sentence to the following sentence from the author response: "For instance, $\beta(\mathbf{x})$ can be the conditional mean of $p(y | \mathbf{x})$."

My point here is that throughout the paper, $\beta$ is used to refer to both a functional of the conditional distribution or a data-based approximation for the said functional. I find the lack of distinction puzzling, if not downright confusing.

2. (The 5th paragraph. I think this is supposed to address my objection about the lack of a precise problem statement.) "The overall goal of the present article is, taking a pre-defined black box $\beta(x)$ that estimates a certain conditional summary statistic of $p(y | \mathbf{x})$, to model $q(y | \mathbf{x})$ under the constraint that if we calculate the summary statistic of this predicted conditional distribution, it will correspond to $\beta(\mathbf{x})$."

- What is $q(y | \mathbf{x})$, and what is its precise relationship to $p(y | \mathbf{x})$? The first occurrence of $q(y | \mathbf{x})$ is followed by the description that it is "a conditional density model," but this can mean many things. Judging from the remainder of the article, $q(y | \mathbf{x})$ is either a model defined through a location-scale family or a model defined via a specification of quantiles. The proposed method is exclusively concerned with the latter kind of models. If this is the case, it would help the reader to have this said explicitly when the symbol $q$ is first introduced.
- The usage of the term "statistic" does not appear to be standard. A _statistic_ is a function of the data $\mathcal{D}$, and is therefore a random quantity. A percentile or a moment of $p(y | \mathbf{x})$, on the other hand, is a functional of the conditional distribution $p(y | \mathbf{x})$, and would be non-random.
- Even ignoring the previous points, the sentence made little sense to me. I will say more on this after the following paragraph.

More on the subject of clarity, notation-wise, I am bothered by the proliferation of $p$'s. Aside from the occurrence in $p(M | \mathbf{x})$, which is rare and hence not a cause for concern, $p$ is used to represent the conditional density (distribution) $p(y | \mathbf{x})$, as well as a generic Chebyshev polynomial approximation $p(\tau,\mathbf{x};d)$. It is possible that I am failing to see some deeper connection here, but if none exists, I would prefer a more clear system of notations.

Returning to the problem statement, after re-reading the paper several times and with the help of the author response, I have arrived at the following interpretation:

Find an estimate / model $q(y | \mathbf{x})$ for the conditional density (distribution) $p(y | \mathbf{x})$ such that $q$ and $p$ have the same $\beta$. Here, $q$ is given by specifying the quantiles, and is approximated by a Chebyshev polynomial of degree $d$. The usual QR methods are inadequate, because the model they output will not have the same conditional percentile / moment / etc. as $\beta$.

However, in situations in which $\beta$ is also being approximated, what is the point in constraining the functional at $\beta$? As pointed out by the authors, if $\beta$ is inaccurate, then this error is propagated to the quantiles, and the performance is worse. If this is the case, why is it of practical interest to enforce the constraint? Had the authors given a concrete, illustrative example, this question may not have arisen. However, absent a practical motivation, it is hard to understand why the constraint needs to be enforced at all.

In any case, suffice it to say that I am still experiencing significant difficulty pinning down the exact goal of the paper.

At the time of writing my initial review, I had hoped that the issues with clarity would be fixed in the revision, leaving me free to evaluate the paper based on the merits of the proposal. However, this has not been the case. Now that I am more familiar with the paper, I realize that the biggest issue about the clarity has much to do with the organization, ambiguous language, etc. Some of it has been addressed in the revision, but the effort did not go far enough.

Even leaving aside any reservation I have about clarity from the point of view of methodology and/or theory, I have to say that I am deeply dissatisfied about the lack of practical motivation. I believe the authors intended the paragraph after Section 2 to be a response to concerns about poor motivation (which have been raised by other reviewers, not I): "The present article was motivated by a real-world need that appears in a pointwise regression forecasting system of a large company. Due to the risk nature of the internal problem where it is applied, uncertainty modeling is important." It would have been nice if the authors had chosen to respond by giving a _concrete_ example, complete with a real data set and an actual task to which their method is an adequate solution. However, the example given in the added paragraph is far too generic and vague to be useful as an illustrative example.

Finally, when I asked for theoretical justification, I was looking for some actual proofs about e.g., consistency guarantees, approximation quality, etc.

---
## Minor
I find the size of the text in the plots to be much too small and hard to read. Also, please label all axes in Figure 3.

---

> ### Author Response · Authors · 2020-11-16
> **Author Response to Reviewer 5**
>
> - R5.Summary.1 [formal description of the problem]
>
> Given a set of samples $\mathcal{D}=( X, Y)=\\{ (\boldsymbol{x}_i, y_i) \mid \boldsymbol{x}_i \in
> \mathbb{R}^D, y_i \in \mathbb{R} \\}_{i=1}^n$, which we assume to be sampled from an unknown distribution $p(y\mid\boldsymbol{x})$. i.e. the real data.
>
> Let us suppose we have a function $\beta \colon \mathbb{R}^{D}\to \mathbb{R}$, which tries to approximate a certain **desired** conditional summary statistic (a percentile or moment) of $p(y \mid \boldsymbol{x})$. For instance, $\beta(x)$ can be the conditional mean of $p(y \mid \boldsymbol{x})$.
>
> The goal of the constrained black-box uncertainty modelling problem is to determine a conditional density model $q(y\mid\boldsymbol{x})$ that approximates $p(y \mid \boldsymbol{x})$, such that the **desired** summary statistic of $q(y\mid\boldsymbol{x})$ corresponds to $\beta(x)$.
>
> In the new Section 2, we emphasise the formal description of the constrained black-box uncertainty modelling problem presented here.
>
> - R5.Summary.2 [no theoretical justification for the proposed method in the paper]
>
> The theoretical justification as well as the real-world motivation to tackle the present problem are now described in Section 2.
>
> - R5.Summary.3 [the proposed method appears to be a heuristic based on approximations that may be reasonable, but requires some justification]
>
> All the proposed methods consider a real constraint (not an approximation) on the relationship between the black-box function, $\beta(x)$, and the different ways to approximate $p(y\mid x)$. To this end, we define $\beta(x)$ and always force $q(y\mid x)$ to satisfy the constraint. Such restriction must be met in any situation, i.e. even if performance worsens because the $\beta(x)$ is not correctly fitted. From this point of view, any worsening in performance should be attributed to the imposed restriction and not to any approximation effect.
>
> - R5.Questions.1 [formal description of the problem the proposed method is trying to address. Also, please give brief but clear descriptions of the existing approaches and their shortcomings.] and Rev5.Questions.4 [Why do N and LP make sense as comparison methods?]
>
> Following Section 2 notation, the constrained black-box uncertainty modelling problem implies considering models that not only approximate the $p(y\mid x)$ with a $q(y\mid x)$, but also constrain $q(y\mid x)$, such that the desired conditional summary statistic (a percentile or a moment) corresponds to $\beta(x)$. At the same time, this $\beta(x)$ will correspond to a previously fixed function that estimates the desired conditional summary statistic of $p(y\mid x)$.
>
> On the whole, not all of the models that approximate $p(y\mid x)$, such as common QR models or a mixture model, can be considered to tackle the present problem. We need models that find a way of disentangling the conditional summary statistic approximation from the rest of the $p(y\mid x)$ modelling. Location-scale family distributions, such as the conditional Normal or conditional Laplace distributions, are directly applicable to this problem, because one of the parameters can be considered as a certain statistic and the other represents information about the confidence. Thus, we considered these two approaches as baselines for comparison.
>
> - R5.Questions.2 [the existing methods do not model the quantiles (or some other functional) of the *output* of some pointwise forecasting system, but rather they model functionals of Y | X.]
>
> We agree that standard QR models estimate the quantiles. However, not in a **constrained way** with respect to a black box, i.e. the desired statistic of the predicted quantiles does not need to satisfy the constraint. Therefore, they cannot be used in the constrained black-box uncertainty modelling context.
>
> - R5.Questions.3 [I do not see why it would be of interest to model the predictions of a black-box forecasting system rather than Y | X.]
>
> Following the Introduction and Section 2, we wanted to model the predictions of a black-box forecasting system in order to maintain the predictions of the black-box function. For instance, because the black box satisfies some interpretability requirements that prevent its replacement while a measure of prediction uncertainty is needed at the same time.
>
> - R5.Questions.5 [Also, there are several instances of errors of grammar / syntax]
>
> We have considered the suggestions made regarding grammar and syntax. The paper is being revised by an English proof-reader to improve its flow.

---

> > ### Comment · AnonReviewer5 · 2020-11-21
> > **Thanks for your response!**
> >
> > Hi authors,
> >
> > I appreciate your detailed response and the changes you mention. I will take another look at the manuscript and update my score in the following days!

---

### Official Review · AnonReviewer3 · 2020-10-28
**Easily readable, but with unclear motivation and utility**

**Rating:** 4
**Confidence:** 2

**Review:**

This paper considers the problem of uncertainty estimation and it proposes to compute quantiles of a black-box predictor using Chebyshev-interpolated quantile regression, where the interpolation parameters are computed with the aid of a neural network.

The paper is generally well-written, though I am leaning toward rejecting the paper because the ideas are poorly motivated and the text does a poor job of convincing readers there is an open problem being addressed here that prior work has not already done. I've provided a few comments and questions below to support my position.

The paper’s clarity could be improved by explicitly describing the research question it considers and motivating its solution choices by connecting them to the research question and related work. Can the authors make these details clear: what is the research question you consider and what contributions does this paper offer?

Currently, it is not clear where this paper sits in relation to prior work. It was mentioned that two starkly different methods -- Dabney et al 2018a and Tagasovska & Lopez-Pax 2018 -- are related to the considered problem but also not applicable, because “the predicted quantiles need to be linked to the black-box prediction in some way.” It seems like such a connection can be established by creating a dataset where the predictions are the targets. Can the authors elaborate on their stance here?

Can the authors please clarify their use of the term “aleatoric uncertainty”? Typically, aleatoric uncertainty describes the inherent variability of a stochastic signal. In contrast, the epistemic uncertainty corresponds to the variability in a stochastic signal’s estimate. This latter type of uncertainty is something that shrinks as more data is collected. The proposed method seems to estimate the uncertainty of a predictor, which would mean it is estimating the epistemic uncertainty of the estimator. However, the paper refers to this as aleatoric. What is the distinction used here?

The proposed approach has questionable utility, because estimating the uncertainty of a prediction is arguably a job for Bayesian methods. The current paper makes very little mention of this and takes for granted that a point estimator is provided, but that one wishes to have uncertainty estimates of the predictions. How does the proposed approach compare to simply computing / estimating a posterior over the prediction? The true posterior will of course be difficult to compute in many domains, but the related work and experimental design should acknowledge the existence of alternatives here: e.g. Bayesian neural nets, variatoinal inference. Can the authors describe when it would be more appropriate to use the proposed method over a Bayesian approach?

Can the authors please justify their use of the UMNN architecture and describe why a simpler alternative, such as standard quantile regression or interpolated quantile regression suffice?

---

> ### Author Response · Authors · 2020-11-16
> **Author Response to Reviewer 4**
>
> - Rev4.1 [the ideas are poorly motivated and the text does a poor job of convincing readers there is an open problem being addressed here that prior work has not already done.] and Rev4.2 [explicitly describing the research question it considers and motivating its solution choices by connecting them to the research question and related work]
>
> To highlight the original real-world need that motivates the present paper, we have created the new Section 2. Together with the Introduction, one of the goals is to remark that, in general, models that approximate $p(y\mid x)$, such as the common QR models or a mixture model, are not able to tackle the constrained black-box uncertainty modelling problem.
>
> - Rev4.3 [what is the research question you consider and what contributions does this paper offer?]
>
> The research question is to build a model that estimates aleatoric uncertainty when a prediction is produced by a black-box function, $\beta(x)$, that represents any statistic of $q(y\mid x)$.
>
> Contribution: To the best of our knowledge, the only models that can tackle this problem are N and LP. We propose the ChePAN, a model that allows estimation of aleatoric uncertainty while forcing any desired statistic to be an external fixed function (the black box). Additionally, aleatoric modelling is done by implicitly approximating the whole distribution of quantiles, which achieve a heterogeneous flexibility in terms of approximating goal distribution (shown in Figure 3).
>
> - Rev4.4 [It seems like such a connection can be established by creating a dataset where the predictions are the targets. Can the authors elaborate on their stance here?]
>
> Following the notation in Section 2.1, these QR models estimate all quantiles of $p(y \mid x)$ in a discrete and implicit way, and predict a $q(y\mid x)$ which does not depend on the black-box prediction $\beta(x)$ in any way, preventing its application to the constrained black-box uncertainty modelling problem.
>
> - Rev4.5 [Can the authors please clarify their use of the term “aleatoric uncertainty”?] and  Rev4.6 [However, the paper refers to this as aleatoric. What is the distinction used here?]
>
> Our use of this term corresponds to the definition of aleatoric and epistemic uncertainties used in Der Kiureghian (2009) and Kendall (2017).
>
> Following Kendall & Gal (2017) and the notation defined in the Introduction, epistemic uncertainty refers to finding the set of models that maximises $p(M \mid \boldsymbol{x})$.
>
> In this work, given that the black box is a single model and its internals are unknown and fixed, we can only estimate $p(y\mid x)$ (aleatoric uncertainty). In other words, we cannot consider a set of black boxes and try to optimise which is better.
>
> - Rev4.7 [The proposed approach has questionable utility, because estimating the uncertainty of a prediction is arguably a job for Bayesian methods]
>
> Following the approach presented by MacKay (1995), Bayesian modelling can be applied to this scenario by considering the response variable a random variable, as we did as a baseline in this article.
>
> On the other hand, the more common Bayesian approach (considering a distribution over the BNN parameters) is not possible since we do not know if the black box is a parametric model. Therefore, this latter Bayesian approach cannot be applied to the presented problem.
>
> - Rev 4.8 [How does the proposed approach compare to simply computing / estimating a posterior over the prediction?]
>
> Estimating $p(y\mid x)$ is not the main aim of this article. We need to link $q(y\mid x)$ to the black-box prediction in some way. Although there are plenty of literature methods that model $p(y\mid x)$, they are not applicable to constrained black-box uncertainty modelling.
>
> - Rev 4.9 [Can the authors describe when it would be more appropriate to use the proposed method over a Bayesian approach?]
>
> Following the notation in the Introduction, if we deal with the constrained black-box scenario, $p(M \mid x)$ cannot be estimated. Thus, the only feasible Bayesian approach is to model $p(y\mid x)$. This paper compares the two approaches (considering a parametric distribution and using the ChePAN).
>
> - Rev 4.10 [Can the authors please justify their use of the UMNN architecture and describe why a simpler alternative, such as standard quantile regression or interpolated quantile regression suffice?]
>
> UMNN considers the derivative wrt the whole input (Section 3). CCN considers the partial derivative wrt only the $\tau$ input (Section A.2). Since the ChePAN avoids the $\tau$-dependence in $C_0$, it can be applied to the constrained black-box uncertainty modelling problem (Section A.3).
>
> Standard Quantile Regression approaches do not enforce a summary statistic provided by an external function (the black box).

---

### Official Review · AnonReviewer1 · 2020-10-28
**A novel method for an important problem**

**Rating:** 6
**Confidence:** 4

**Review:**

This paper presents an approach to model aleatoric uncertainty for the black-box model, through the combination of Chebyshev polynomial approximation and quantile loss. The method seems novel to the prior work and achieves good results in capturing uncertainty in predictive modeling. My detailed comments are as follows:

Strength:
1. By constraint the neural network model for the derivative function $\phi(\tau, x)$ to be strictly positive, this method solves the quantile crossing issue which is a common problem in simultaneous quantile regression.
2. This method does not assume any prior knowledge of the model structure, which means one doesn't need to modify the parameters of the original model, and therefore separates the constraint from the model training procedure.

Weakness:
From section 4 and Appendix it looks like the way to compute each coefficient is in a sequential manner, which may not be efficient enough.

Additional Questions:
1. If ChePAN is a separate module of the black-box model, is this similar to a post-processing method for pre-trained models to generate user-specified uncertainty intervals?
2. Following the previous question, if that is the case, does that mean we need to train the black-box model to produce $\phi(\tau, x)$ for later use? Or add an additional neural network called uncertainty wrapper for that?
3. It looks like CCN is not discussed in the experiment section.

UPDATE after author response:
Estimating arbitrary quantiles to approximate the predictive distribution of a black-box model is a novel problem to look into. In response, the authors have attempted to address my major concern and they are mostly clear. I would suggest the authors add more discussion on the motivation of the problem settings, e.g., whether it arises from an actual business problem, why satisfying constrained black-box uncertainty problem is a must-have, etc.

---

> ### Author Response · Authors · 2020-11-16
> **Author Response to Reviewer 3**
>
> - R3.weakness.1 [the way to compute each coefficient is in a sequential manner, which may not be efficient enough]
>
> Although we are performing DCT-II to reduce the Eq. 17 calculation to $\Theta(d \log d)$, we agree that computing each coefficient to build a Chebyshev polynomial to then integrate it to obtain $P$ in Eq. 7 is less efficient than a model that is directly optimised using QR loss (Eq. 1). However, in spite of the fact that these models implicitly predict the whole quantile distribution $q(y\mid x)$, we cannot force their predicted quantiles to satisfy the constraint as required by the constrained black-box uncertainty problem. Therefore, they cannot be used to solve the main problem of the article.
>
> On the other hand, conditional Normal or Laplace distributions can be used in this problem (as in the Experiment section). However, the conditional summary statistic that constrains the forecasted $p(y\mid x)$ is always the same: the mean in the case of the Normal and the median in the case of the Laplace distribution. Thus, freedom in the choice of the constraint (described in Section 5.1), and flexibility in the approximation of $p(y\mid x)$, by using QR, are two advantages to consider of the ChePAN approach.
>
> - R3.AQ.1 [If ChePAN is a separate module of the black-box model, is this similar to a post-processing method for pre-trained models to generate user-specified uncertainty intervals?]
>
> Yes, as long as the intervals to be predicted satisfy a pre-established relationship with the black-box forecast (i.e. the so-called constraint of the constrained black-box uncertainty modelling scenario). However, as far as we know, the generation of uncertainty intervals is typically defined regardless of the value of a pre-trained predictor or, at best, without exactly specifying the relationship between the predictor and the extremes of the predicted interval. This is the main limitation when applying these models in the comparison provided. Additionally, the goal of the ChePAN is to provide not only certain pre-specified quantiles to generate an interval, but to obtain all the quantiles $\tau \in [0,1]$, such that $ p(y\mid x)$ can be inferred in a discrete way.
>
> - R3.AQ.2 [if that is the case, does that mean we need to train the black-box model to produce for later use?]
>
> In the main scenario of the article, the black box is given and fixed. We cannot therefore consider it being "re-trained", since this black box could even be a regression system based on deterministic rules (as shown in Figure 1). As explained in the Introduction and Research goal sections, there is a variety of reasons in real-world scenarios where this restriction is necessary; to satisfy appropriate regulations or interpretability constraints, for instance.
>
> - R3.AQ.3 [It looks like CCN is not discussed in the experiment section]
>
> Similarly to UMNN, CCN cannot be used in the main problem of the article: constrained black-box uncertainty modelling. Unlike the UMNN, the CNN was proposed in this article as a natural progression from a model in the literature to estimate the derivative (UMNN) and fulfil the main goal of the article (to create the ChePAN), which estimates the partial derivative (see Sections 4 and 5). However, since the $C_0$ of the CCN depends on $\tau$, it cannot be applied to the general constrained black-box uncertainty modelling scenario. This is why it cannot be included in the experimental comparison. To highlight this, we have decided to leave the introduction to the CCN and its comparison with the ChePAN for Sections A.2 and A.3 of the Appendix and focus the explanation in Section 5 on the ChePAN model.

---

### Official Review · AnonReviewer4 · 2020-10-31
**Neat approach to black-box uncertainty modeling**

**Rating:** 7
**Confidence:** 4

**Review:**

This paper proposes a novel approach to modeling uncertainty, as an layer added-on to an otherwise black-box system. The ChePAN uses a neural network to estimate per-quantile roots of a chebyshev polynomial, then uses a quantile regression loss to fit these coefficients using backpropagation. Importantly, the Chebyshev polynomial formulation gives the practitioner some flexibility around deciding which statistic of the conditional CDF should be matched by the black box system. Examples are given for matching the 0 and 1 quantiles, for cases of min and max estimation, as well as matching either of median or mean.

The authors make a case for why previous/related approaches to black box uncertainty modeling have deficiencies e.g. the roots considered by the Clenshaw-Curtis quadrature extension of UMNN (CCN) are bounded by the quantile being considered, which is somehow limiting, though I was a little hazy on why.
I would be curious whether the authors have considered splines as another alternative, at least for modeling black boxes known to be in a bounded, e.g. 0-1, range. https://arxiv.org/abs/1906.04032 is perhaps relevant there.

The idea is quite a neat one. One uses a neural network to emit positive-valued roots of a Chebyshev polynomial, which are optimal function approximators on the 0-1 interval, perfect for a flexible CDF inversion, thus an extremely flexible distribution estimator. This is what enables setting a statistic of choice equal to the output of the black-box system. The actual implementation does not seem overly complex, which should make it widely accessible to practitioners. The idea of combining quantile regression with deep networks for fitting conditional distributions is not new, as the authors acknowledge, but this work provides a mechanism for matching a statistic to that distribution, while also being able to fit that distribution using a QR loss.

The experiments use datasets comparable to prior/related work. However, tables and plots don't seem to include for comparison reference points like Brando, 2019's UMAL which appears to provide a nicely calibrated estimator for the AirBnB datasets. For example, I would prefer to see the calibration plot on the AirBnB datasets in Fig 4 left side, for comparison with this earlier work. Since the baselines are all internally sourced it's hard to tell if this paper is achieving state of the art calibration, or only providing a novel approach to doing so. On the topic of baselines not particularly comparable to prior work: the "N" and "LP" baselines, which in prior work often use neural networks for both the mean and the scales, in this work use RF/XGBOOST for the mean. It would be good to understand whether using a NN for the mean estimator yields changed calibration metrics (thus changing the baseline competitiveness).

On the whole, I think this is an exciting new contribution and would recommend accepting, but I think the paper would benefit from clearer evidence that the approach provides a value-add relative to existing techniques. If I'm meant to understand that "N" and "LP" baselines are that evidence, I must have misunderstood, because to me those look like custom baselines built for this work.

---

> ### Author Response · Authors · 2020-11-16
> **Author Response to Reviewer 2**
>
> - R2.1 [the roots considered by the Clenshaw-Curtis quadrature extension of UMNN (CCN) are bounded by the quantile being considered, which is somehow limiting, though I was a little hazy on why]
>
> Comparing Eq. 20 (of the CCN) and Eq. 7 (of the ChePAN), we observe that:
>
> -- Firstly, the CCN defines its roots in the interval $[0,\tau]$ (Eq. 18). This implies obtaining Chebyshev coefficients that are $\tau$-dependent (Eq. 19). Consequently, the definition of $C_0$ depends on $\tau$, and $\beta(x)$ can therefore only represent the lower quantile, $q=0$ (Eq. 20).
>
> -- Secondly, the ChePAN defines its roots in a fixed interval $[0,1]$ (Eq. 3). In contrast to the CCN, the Chebyshev coefficients are $\tau$-independent (Eq. 7). Therefore, the freedom of $C_0(x)$ allows us to impose a constraint for the whole interval, which makes it possible to define different types of constraints (see Section 5.1).
>
> In the new version of the article, we have extended the introduction of the CCN model (see Section A.2) and the comparison between the ChePAN and CCN (see Section A.3) to include a better description of this point.
>
> - R2.2 [the authors have considered splines as another alternative, at least for modeling black boxes known to be in a bounded]
>
> Splines are definitely something to look into as a future work. The Chebyshev approach exhibits uniform properties for the whole interval, while splines show local properties guaranteed with fewer regularity assumptions. Splines are better at capturing spikes. Generically, Chebyshev has better convergence and either splines or splitting Chebyshev can be considered for spike detection. On the other hand, neural spline flows contain a monotonic rational-quadratic transformation that could be extended to the current context. However, this would comprise a new line of research that we have not yet developed.
>
> - R2.3 [I would prefer to see the calibration plot on the AirBnB datasets in Fig 4 left side, for comparison with this earlier work]
>
> The UMAL model (similarly to other mixture models) cannot be compared in the context of constrained black-box uncertainty modelling, because it is not possible to disentangle the uncertainty estimation of $p(y \mid x)$ from a certain pointwise predictive system (the black-box $\beta(x)$) when a mixture of distributions is considered. Therefore, we agree that the calibration plot can be compared with respect to which model obtains best calibration performance but, ultimately, the UMAL cannot be applied to a solution for this problem. Thus, since it could not be used to solve our main problem in the article, we have not considered it for comparison.
>
> - R2.4 [It would be good to understand whether using a NN for the mean estimator yields changed calibration metrics (thus changing the baseline competitiveness)]
>
> We completely agree that this analysis will be interesting in a longer version of the current article. However, here we show two results that could indicate good performance in this regard:
>
> -- At a quantitative level, in the last 3 rows of Table 1, we find the value of the QR loss obtained using a neural network as a black box. QR loss, as defined, gives us information on the quality of the predicted quantiles.
>
> -- On the other hand, at a qualitative level, Figure 6 shows the performance of estimating the quantiles with different types of black boxes (and, where each black box is a neural network).
>
> - R2.5 [the paper would benefit from clearer evidence that the approach provides a value-add relative to existing techniques]
>
> Location-scale family (specifically, the normal and Laplace distributions) have a special property that allows us to disentangle the estimation of a certain statistic with respect to $p(y \mid x)$ and the uncertainty estimation. This property is not common in models such as typical QR models, mixtures and other alternatives to model $p(y\mid x)$. Therefore, although the conditional normal and Laplace distributions are very simple models, they were the only alternatives we found to apply to the constrained black-box uncertainty modelling problem. To the best of our knowledge, the ChePAN represents the first alternative to these simple models in the constrained black-box scenario. Additionally, it makes two important novel contributions compared to the aforementioned:
>
> -- Firstly, it allows us to pre-decide the type of summary statistic constraint that will be defined using the $\beta(x)$ (as shown in Figures 1 and 2).
>
> -- Secondly, it can learn a heterogeneous distribution $q(y\mid x)$ that approximates $p(y\mid x)$ because it implicitly approximates all the quantile values (as we can see in Figures 3 and 6).

---

### Official Review · AnonReviewer5 · 2020-11-06
**This paper contains good contributions but is held back by poor presentation.**

**Rating:** 7
**Confidence:** 2

**Review:**

-------------------
# Update

After reading the updated version of the paper, I think my main issues with presentation have been addressed. I now see this work as introducing a relatively niche problem and solving it elegantly. The text is clear enough for an average ICLR attendee to understand and contains additional background information in the appendix. I have increased my score to a 7.

-------------------

### Summary / Contributions

This paper adapts quantile regression with NNs to settings in which a summary statistic is fixed (some percentile or moment). This is done by having a NN predict the coefficients of a Chebyshev polynomial which is then integrated to evaluate the predicted cumulative density function over targets. Freedom in the choice of the constant of integration is used to enforce a summary statistic provided by a black box model (BBM).

The method is used to provide heteroschedastic noise models for black box regression systems and validated on standard regression benchmarks (UCI).

### Originality and Significance

To the best of my knowledge, the novel aspects of this work are:

* The development of a derivative based approach to quartile regression which allows for the output distribution to be conditioned on an input x. This could be used very generally as a heteroscedastic noise model.
* The extension of the above approach to allow enforcing some summary statistic like distribution mean or percentiles.
* The introduction of post-hoc noise modelling for black-box decision making systems as a task.

#### Pros:
* Sound solution to a (to the best of my knowledge) new task: constrained heteroscedastic noise modelling with free-form / implicit distributions.
* Good empirical performance on benchmark datasets.

#### Cons
* The paper is not very well written and hard to follow. I had to read it multiple times to understand it well. I felt important details where glossed over or missing in the provided explanations.
* I would have liked to see the method being evaluated further by using it with a wider variety of black box models and different amounts of training data.
* Motivation of adding noise models to black box predictors is a bit weak considering no evaluation is performed on real-world scenarios with BB models in deployment.

### Other Comments:

#### Method:

This method is completely agnostic to the BBM and, as such, its uncertainty might be in contradiction with the BBM given statistic:
When a noise model is learnt in conjunction with a predictive model (e.g. a NN that predicts mean and st deviation values for each input) , aleatoric uncertainty usually reflects what the model can’t predict. In the limited data regime, this is often a crude approximation to the inherent uncertainty in the generative process of the data. A not very flexible (e.g. linear) model will see non-linearity in the function being learnt as noise in the data. When using a very flexible model (e.g. NN) might be able to fit all datapoint and not see anything as noise.  As such, I would have liked to see more exhaustive evaluation of the proposed methods in the following settings:

*  It would be interesting to see the effectiveness of this technique on varying sized datasets when the NN used for quantile prediction is very flexible.
*  What happens when the BBM is not noisy (as in Fig 3.) but instead biased because it is not flexible enough for problem at hand. This his often the case in real world deployments when the model is chosen for interpretability (e.g. decision tree or linear regressor) — I imagine the inability of this approach to consider model specification uncertainty, could cause noise distributions to misbehave when the BBM is very badly specified.


#### Experiments

The provided baseline methods seem relatively weak: just unimodal heteroscedastic noise models. However, I do not know of any existing approaches that impose summary statistic constraints on arbitrarily complex predictive distributions.

#### Presentation
* The language used is a bit ungrammatical at times. I would recommend going over the paper with a grammar checker. There are some paragraphs that are only 1 sentence and feel a bit disconnected.
* A brief overview of key preliminaries would be convenient: Relevant properties of Chebyshev polynomials, how the DCT can be used to obtain coefficients, etc
* Some important concepts are glossed over or referenced to succeeding sections, despite them being necessary to follow the paper well. It is better to reference previous sections than future ones.
	* It was not clear to me from the explanation at the end of section 2 why existing quantile regression methods would not be applicable to the constrained black box scenario.
	* It was not clear to me from section 3 why the solution to the above problem required using a UMNN style model.
	* In Section 4 it is not clear what “Due to the aforementioned truncation in ChePAN” refers to, as above you state both CCN and ChePAN calculate coefficients of truncated expansions. In general, it is not very clear to me how mentioning CCN in section 4 is helpful to the reader. I would recommend just explaining ChePAN in more detail and leaving CCN for the appendix.
	* The reference to section 4.1 in the paragraph after equation 3 leaves me wanting an explanation of why we want Chebyshev roots to fit in [0,1] instead of [0, \tau].
* What does the bolding in Figure / Table 4 refer to? It seems inconsistent. In table 1 it indicates best methods.

### Summary
I think this paper contains good contributions but is held back by poor presentation. If this is corrected by the authors, I would be happy to recommend acceptance.

---

> ### Author Response · Authors · 2020-11-16
> **Author Response to Reviewer 1**
>
> - R1.C.1 [hard to follow] and R1.P.1 [grammar checker]
>
> The text is being revised by a professional English proofreader. Regarding the suggestion made, the whole text is being revised for typos and grammar.
>
> - R1.C.3 [no evaluation is performed on real-world scenarios with BB models in deployment]:
>
> Although our main motivation for tackling Constrained Black-Box Uncertainty Modelling (**CBBUM**) arose from a real-world need, which we have highlighted in the new Section 2, the privacy and security concerns regarding this real-world problem have led us to perform the comparison using publicly-known data sets such as the UCI and the Airbnb data set.
>
> * R1.M.1 [its uncertainty might be in contradiction with the BBM given statistic]:
>
> You are absolutely right, our aleatoric uncertainty has two sources:
>
> (1) The real data generation process
>
> (2) The mismatch between the black box and the real statistic (observed in Figure 3).
>
> We clarify this point in Section 2. In fact, type (2) can be seen in the upper part of Figure 3, where the “poorly approximated RF” propagates its noise to the predicted quantiles (in blue) because the constraint is always forced.
>
> * R1.M.2 [When using a very flexible model (e.g. NN) might be able to fit all datapoints and not see anything as noise]:
>
> Following the two aleatoric uncertainty sources described above, using a very flexible model for $\beta(x)$ might fit the real statistic, which would mean that the aleatoric uncertainty is originated only by the data. Note that standard regression NNs are optimised to learn a certain statistic (e.g. when the loss is the MSE, then the learnt statistic is the mean, whereas with the MAE it is the median).
>
> * R1.M.3 [Effectiveness of this technique on varying sized datasets]
>
> We will take this improvement into account for an extension of this work we are preparing. On the other hand, Table A.7 shows that the UCI data sets used have different sizes and the performance of all the compared models is usually preserved.
>
> * R1.M.4 [What happens when the BBM is not noisy (as in Fig 3.) but instead biased]
>
> Given $\beta(x) = s(x)+b$, where $s(x)$ is a certain real conditional statistic and $b$ a fixed constant bias, forcing $\beta(x)=NN(x)$ to be that statistic of $q(y\mid x)$ in the ChePAN is similar to considering a $\mathcal{N}(\beta(x), \sigma(x))$ and estimating the $\sigma$ function: the biased approximation of $\beta$ will affect $\sigma$, but $\beta$ will be that statistic of $q(y\mid x)$, as desired (see Section 2).
>
> * R1.E [The provided baseline methods seems relatively week]
>
> We agree, which is why we use these baselines. N and LP (i.e. location-scale family) have one parameter that can be considered a certain statistic and another that give us information about the confidence. This is not a common property in models that estimate $p(y\mid x)$. The ChePAN makes an important new contribution compared to such models, because it can pre-establish the type of summary statistic constraint.
>
> * R1.P.2 [A brief overview of key preliminaries would be convenient]
>
> We have added a new Appendix - A.1 - to summarise the Chebyshev polynomials.
>
> * R1.P.3 [why existing quantile regression methods would not be applicable]
>
> Standard QR models only approximate the desired quantiles to produce $q(y\mid x)$ but do not satisfy the constraint wrt the black-box model. Therefore, QR optimisation is not enough to tackle the constrained black-box uncertainty modelling scenario (as shown in Section 3).
>
> * R1.P.4 [why the solution to the above problem required using a UMNN style model]
>
> The UMNN cannot be used in CBBUM. We have only introduced it to connect with other methods in the literature. It is a method that computes the derivative of the loss function wrt all the input. The CCN is our extension for computing the partial derivative wrt only the $\tau$ input, also not generically applicable to CBBUM. Finally, the ChePAN allows us to "free up" the constant of integration wrt $\tau$ so that we can tackle CBBUM.
>
> * R1.P.5 [what "Due to the aforementioned truncation in ChePAN" refers to] and R1.P.7 [why we want Chebyshev roots to fit in [0,1] instead of [0, \tau]]
>
> The $[0,1]$ is convenient when considering any desired black box (see Section 5.1). In the case of $[0,\tau]$, only a black box for the 0th quantile is possible (see new Sections A.2 and A.3).
>
> * R1.P.6 [leaving CCN for the appendix]
>
> We have followed the recommendation and moved the CCN to Section A.1, where it is introduced and then compared to the ChePAN.
>
> - R1.P.8 [what does the bolding in Figure/Table 4 refer to?]
>
> Values in bold have ranges that match the best result. We have added this clarification to the figure description.
>
> - R1.S.1 [poor presentation. If this is corrected by the authors, I would be happy to recommend acceptance.]
>
> We are uploading a new version of the paper with your suggestions addressed.

---

### Author Response · Authors · 2020-11-16
**General comment and notation**

Dear Reviewers,

Thank you so much for reviewing our paper. We believe that all your comments and suggestions have been taken into account in the new version and we also hope that our personalised response will be clear enough.

Based on common suggestions, we decided to introduce some new sections and make minor changes in the others. The new Section 2 addresses the need for a clear objective and contribution of the present work. Specifically, we have stressed that our motivation comes from a real-world problem in a large company and our main contribution is to propose a flexible solution for constrained black-box uncertainty modelling.

Another common observation was the need for a clear distinction between the UMNN, CCN and ChePAN. Since the main contribution is based on the ChePAN approach, following Rev. 1's suggestion, we finally decided to make Section 5 an exclusive description of the ChePAN and refer the CCN to a special section in the Appendix: A.2. Moreover, we have made minor changes to Section 4 in order to refer to the new information about the CCN in the Appendix.

Rev. 1 rightly suggested that a preliminary explanation of basics of Chebyshev polynomials should be included to help the reader follow some of the steps in Section 5. We have therefore added a new Section - A.1 - to the Appendix, which summarises the main properties of Chebyshev polynomials to help the reader easily follow the main sections of the paper.

In response to the reviewers' questions, we have established a notation before each answer to label them:

* R x . s . n [ q ]

where "x" is the number of the reviewer, "s" is the section name where the question was raised (if any) or the initial of the section, e.g. Presentation->P, "n" is the question number and, finally, "q" is some part of the original question to identify what we are responding to.

---

> ### Author Response · Authors · 2020-11-19
> **New revised version of the article**
>
> We have now uploaded the version of the article revised by a professional English proofreader.
>
> We believe that all of the Reviewers' comments and suggestions have been taken into account and would like to express our sincere thanks to all of them for their efforts.

---

> ### Author Response · Authors · 2020-11-24
> **Review of paper ChePAN: Constrained Black-Box Uncertainty Modelling with Quantile Regression**
>
> Dear reviewers,
>
> We hope that our feedback and amendments to the paper address all of your comments and you will therefore reconsider it for acceptance.
>
> We have followed all of your recommendations to update the paper. Specifically, these changes include: a thorough revision by an English proofreader to improve its flow, connecting each reviewer's question with the results and text, adding a new Appendix (A.1) to summarize Chebyshev polynomials, and moving the introduction of the CCN and its comparison with the ChePAN to the Appendix section.
>
> We believe that your comments have helped us to significantly improve the final version of the paper.
>
> Thank you again for your time and helpful comments,
>
> Best regards
>
> The authors

---

### Decision · Program_Chairs · 2021-01-07
**Final Decision**

**Decision:**

Reject

**Comment:**

The paper proposes to model uncertainty by combining quantile
regression and Chebyshev polynomial approximation. The paper addresses
the important problem of uncertainty quantification for black box
models. However, some major concerns remain after the discussion among
the reviewers. In particular, there has been some concerns around the
clarity of the presentation. The proposal lacks a clear use case, e.g.
where satisfying constrained black-box uncertainty problem is a
must-have.